

# Identification of candidate genes and prognostic value analysis in patients with PDL1-positive and PDL1-negative lung adenocarcinoma

Xiaoguang Qi[1], Chunyan Qi[2], Xindan Kang[1], Yi Hu[1] and Weidong Han[3]

[1] Department of Oncology, Chinese PLA General Hospital, Beijing, China
[2] Department of Special Ward, Chinese PLA General Hospital, Beijing, China
[3] Department of Bio-therapeutic, Chinese PLA General Hospital, Beijing, China

## ABSTRACT

**Background**. Increasing bodies of evidence reveal that targeting a programmed cell death protein 1 (PD-1) monoclonal antibody is a promising immunotherapy for lung adenocarcinoma. Although PD receptor ligand 1 (PDL1) expression is widely recognized as the most powerful predictive biomarker for anti-PD-1 therapy, its regulatory mechanisms in lung adenocarcinoma remain unclear. Therefore, we conducted this study to explore differentially expressed genes (DEGs) and elucidate the regulatory mechanism of PDL1 in lung adenocarcinoma.

**Methods**. The GSE99995 data set was obtained from the Gene Expression Omnibus (GEO) database. Patients with and without PDL1 expression were divided into PDL1-positive and PDL1-negative groups, respectively. DEGs were screened using R. The Gene Ontology (GO) database and Kyoto Encyclopedia of Genes and Genomes (KEGG) were analyzed using the Database for Annotation, Visualization and Integrated Discovery. Protein–protein interaction (PPI) networks of DEGs was visualized using Cytoscape, and the MNC algorithm was applied to screen hub genes. A survival analysis involving Gene Expression Profiling Interactive Analysis was used to verify the GEO results. Mutation characteristics of the hub genes were further analyzed in a combined study of five datasets in The Cancer Genome Atlas (TCGA) database.

**Results**. In total, 869 DEGs were identified, 387 in the PDL1-positive group and 482 in the PDL1-negative group. GO and KEGG analysis results of the PDL1-positive group mainly exhibited enrichment of biological processes and pathways related to cell adhesion and the peroxisome proliferators-activated receptors (PPAR) signaling pathway, whereas biological process and pathways associated with cell division and repair were mainly enriched in the PDL1-negative group. The top 10 hub genes were screened during the PPI network analysis. Notably, survival analysis revealed *BRCA1*, mainly involved in cell cycle and DNA damage responses, to be a novel prognostic indicator in lung adenocarcinoma. Moreover, the prognosis of patients with different forms of lung adenocarcinoma was associated with differences in mutations and pathways in potential hub genes.

**Conclusions**. PDL1-positive lung adenocarcinoma and PDL1-negative lung adenocarcinoma might be different subtypes of lung adenocarcinoma. The hub genes might play an important role in PDL1 regulatory pathways. Further studies on hub genes are warranted to reveal new mechanisms underlying the regulation of PDL1 expression.

Corresponding author
Yi Hu, huyi0401@aliyun.com

These results are crucial for understanding and applying precision immunotherapy for lung adenocarcinoma.

## INTRODUCTION

Non–small-cell lung cancer (NSCLC) is a leading malignancy threatening human life and health worldwide (*Herbst, Morgensztern & Boshoff, 2018*; *Zhang et al., 2019*). Lung adenocarcinoma is the most common type of NSCLC, and it is a more heterogeneous subtype of NSCLC compared with squamous cell carcinoma. Therefore, its individualized treatment and prognosis have attracted considerable attention; understanding its biological characteristics is necessary for achieving such individualized treatment and prognosis. In the past decade, studies have identified tyrosine kinase inhibitors (TKIs) targeting epidermal growth factor receptor (EGFR), anaplastic lymphoma kinase (ALK), and ROS proto-oncogene 1 (ROS1) as potential therapies for lung adenocarcinoma on the basis of genotyping (*Sgambato et al., 2018*; *Singhi et al., 2019*). Molecular targeted therapy based on these sensitive targets has considerably enhanced overall survival (OS) in lung adenocarcinoma. Despite the progress in the treatment of lung adenocarcinoma, the mortality rate among patients with advanced lung adenocarcinoma remains high, with the 5-year survival rate being approximately 15% in patients with advanced NSCLC (*Blandin Knight et al., 2017*). Nevertheless, a breakthrough was achieved in terms of immunotherapy for cancer according to recent reports, with immune checkpoint inhibitors prolonging survival in some patients with various cancer types (*Yu & Wang, 2018*; *Lorigan & Eggermont, 2019*; *Morse, Hochster & Benson, 2020*). In particular, targeting programmed cell death protein 1 (PD-1) monoclonal antibodies is rapidly becoming a promising therapeutic approach for NSCLC treatment (*El-Osta & Jafri, 2019*). Patients with >50% PDL1 expression treated using anti-PD-1 therapy showed a longer progression-free survival (PFS) period than did those treated using standard chemotherapy as the first-line treatment for NSCLC. Furthermore, in the past few years, some of these antibodies have been successfully commercialized and approved as first- and second-line treatments for advanced NSCLC (*Gridelli et al., 2018*; *Dafni et al., 2019*). In particular, immunotherapy is different from traditional chemotherapy or targeted therapy in that it has durable clinical benefits and fewer side effects in general (*Suresh et al., 2018*). Hence, immunotherapy is a promising therapy for lung adenocarcinoma; however, its application is limited to a subtype of patients with lung adenocarcinoma. Individualized immunotherapy guided by predictive biomarkers is crucial for improving the prognosis of lung adenocarcinoma.

Recent studies have proposed PDL1 expression, tumor mutational burden, and DNA mismatch repair deficiency as biomarkers for anti-PD-1 therapy (*Teng et al., 2018*; *Darvin et al., 2018*; *El-Osta & Jafri, 2019*). In particular, PDL1 expression—recommended by the

National Comprehensive Cancer Network guidelines—is widely recognized as the most powerful predictive marker for immunotherapy in lung adenocarcinoma. Moreover, patients with PDL1-positive lung adenocarcinoma are more likely to benefit from immunotherapy than those with PDL1-negative lung adenocarcinoma (*Gridelli et al., 2018*; *Dafni et al., 2019*), suggesting that PDL1 expression plays a major role in the pathogenesis of lung adenocarcinoma. However, the mechanism underlying this clinical problem remains unclear. Understanding the regulatory role of PDL1 expression in lung adenocarcinoma is crucial for precision immunotherapy. Currently, PDL1 expression is primarily determined using immunohistochemistry (IHC) assays; nevertheless, IHC assays yield inconsistent results due to variable cutoffs and different antibodies with differing affinities (*Hunter, Socinski & Villaruz, 2018*). Hence, such assays cannot comprehensively elucidate the molecular mechanism and enriched pathways underlying the efficacy of immunotherapy. To solve this problem, novel strategies are imperative to explore the intrinsic mechanism of PDL1 expression in the biology of lung adenocarcinoma.

Because of advances in next-generation sequencing technology, high-throughput sequencing results on cancers have been increasingly released on the Gene Expression Omnibus (GEO), a public database; this can thus enable understanding the molecular mechanism of PDL1 expression and pathogenesis of lung adenocarcinoma. Using next-generation sequencing technology, previous studies have successfully identified key biomarkers in lung adenocarcinoma, which has proved to be essential in understanding the molecular mechanism of tumors (*Tang et al., 2018*; *Li et al., 2019*). Because PDL1 expression might affect certain key pathways in immunotherapy for lung adenocarcinoma, exploring the differences between patients with PDL1-positive and those with PDL1-negative lung adenocarcinoma would provide insights into the regulatory mechanisms of immunotherapy. Hence, it is reasonable to explore the differences in gene expression profiles and enriched pathways between the two groups of patients. Using the GEO database, we studied a gene expression dataset (GEO accession number: GSE99995) and comparatively analyzed sequencing data of gene expression between patients with PDL1-positive and PDL1-negative lung adenocarcinoma using the Agilent oligonucleotide microarray system. The aim of the study was to explore the differences in gene expression profiles and enriched pathways between the two groups of patients and identify the potential biomarkers predicting patient prognosis in order to elucidate the regulatory mechanism of PDL1 in immunotherapy, which could be crucial in guiding precision immunotherapy and improve prognosis in lung adenocarcinoma.

## MATERIAL AND METHODS

### Gene expression profile data

The GEO database is a public functional genomics data repository (https://www.ncbi.nlm.nih.gov/geo/). We obtained gene expression profile data from the public GEO database based on the keywords "PDL1 expression", "lung adenocarcinoma", and "homo sapiens". GSE99995 was retrieved. To prevent interferon expression levels having effects on PDL1 gene expression profiles, only patients with lung adenocarcinoma with low interferon

expression levels were included. Patients with and without PDL1 expression were divided into PDL1-positive (3 people, average age: $52.33 \pm 4.04$ years, tumor stage IIIA) and PDL1-negative (3 people, average age: $51.33 \pm 3.06$ years, tumor stage IIIA) groups, respectively.

### DEGs analysis and mapping

A series of matrix files and platforms were downloaded and processed using R. Screening of differentially expressed genes (DEGs) between the two groups as well as heat mapping and volcano mapping were performed using Limma. A log FoldChange of $>2$ and an adjusted $P$ value of $<0.05$ indicated the presence of DEGs.

### Gene ontology and kyoto encyclopedia of genes and genomes pathway enrichment analysis

We used Gene Ontology (GO) and Kyoto Encyclopedia of Genes and Genomes (KEGG) pathway enrichment analyses to understand gene functional annotation and functional enrichment, respectively. We used the Database for Annotation, Visualization and Integrated Discovery (DAVID) to perform GO and KEGG annotation of DEGs. DAVID is an online database (https://david.ncifcrf.gov/) for estimating functional domains and biological implications. Fisher's exact test was employed for analyses of pathways, diseases, and functions. A $P$ value of $<0.05$ is recommended because it denotes the significance of GO terms and KEGG pathway enrichment in genes. The top 10 GO terms and KEGG pathway enrichment results were mapped using Hmisc and ggplot2 in R.

### Gene set enrichment analysis pathway enrichment analysis and validation

Gene Set Enrichment Analysis (GSEA) is a computational method—based on the analysis of all genes—that determines whether an a priori defined set of genes shows statistically significant differences between two biological states. In the KEGG pathway analysis, the threshold value was set according to the expression level, which could only be analyzed in genes with significantly different expression levels. To avoid limitations in our results, GSEA was used again to consider the effect of all DEGs, not limited to those with significantly different expression levels. Therefore, GSEA was used to comprehensively analyze the differences in gene pathway enrichment results between the two groups.

### Protein–protein interaction network and analysis of hub genes

STRING is an online database designed to evaluate and predict protein–protein interactions (PPIs) (https://string-db.org/cgi/input.pl). First, STRING was used in this study to analyze the PPI network of different genes in the two patient groups. Isolated nodes were removed, and the results of the interaction network were downloaded and then imported into Cytoscape (version 3.7.2) for subsequent analysis. Second, the PPI network was constructed and visualized using Cytoscape (version 3.7.2) and cytoHubba. The MNC algorithm was employed to screen and identify the hub genes that might be key candidate genes with crucial regulatory functions.

### Clinical characteristics and survival analysis related to hub genes

Gene Expression Profiling Interactive Analysis (GEPIA) is an online cancer data mining site that is based on the TCGA and GTEx databases and uses a standard processing pipeline (http://gepia.cancer-pku.cn/detail.php). In this study, the online survival analysis tool of GEPIA was used to verify our previous results obtained using the TCGA and GTEx databases. Furthermore, differences in the expression of hub genes in lung adenocarcinoma and adjacent normal tissues were analyzed. Moreover, the relationships among differences in expression of hub genes, pathological staging, and prognosis in lung adenocarcinoma were analyzed. Furthermore, $P$ values of $<0.05$ were considered statistically significant.

### Mutation characteristics and pathway analyses of hub genes

We investigated the mutation characteristics and possible functional mechanisms of the differential expression of hub genes in lung adenocarcinoma. Therefore, data sets containing gene expression profiles of patients with lung adenocarcinoma were included in the TCGA database (e.g., Broad, Cell 2012; MSKCC, Science (2015); TCGA, Firehose Legacy; TCGA, Nature (2014); and TCGA, PanCancer Atlas). A combined study of five data sets including 1598 patients and 1600 samples were included in the analysis.

GSCALite is a web-based analysis platform that analyzes an entire set of genes in cancers (http://bioinfo.life.hust.edu.cn/web/GSCALite/), including alterations in DNA or RNA of cancer-related genes, the activity of 10 cancer-related pathways, and miRNA regulatory network for genes. To understand the cancer-related pathways of hub genes, we distinguished hub gene expression between pathway activation and inhibition groups according to pathway scores in GSCALite.

## RESULTS

### Identification and mapping of DEGs

Our results revealed that the PDL1-positive and PDL1-negative groups comprised 34,729 genes. According to criteria of a log FoldChange of $>2$, and an adjusted $P$ value of $<0.05$, a total of 869 DEGs were identified, with 387 and 482 DEGs in the PDL1-positive and PDL1-negative groups, respectively. The corresponding heat map and volcano map are shown in Fig. 1.

### GO terms and KEGG pathway enrichment analysis of DEGs

The analysis of DEGs by using GO terms was divided into three parts, namely biological processes, cellular components, and molecular functions. The GO analysis of the PDL1-positive group mainly revealed the enrichment of pathways involved in cell adhesion (BP, GO:0007155) plasma membrane (CC, GO:0005886), and cadmium ion binding (MF, GO:0046870). The GO analysis of the PDL1-negative group mainly revealed the enrichment of pathways involved in cell division (BP, GO:0051301), nucleoplasm (CC, GO:0005654), and DNA binding (MF, GO:0003677). The detailed results are shown in Table 1. The KEGG pathway enrichment results in the PDL1-positive group revealed that the enriched pathways were mainly those involved in cell adhesion molecules, PPAR signaling, and ECM-receptor interaction. However, the KEGG pathway enrichment results of the PDL1-negative group
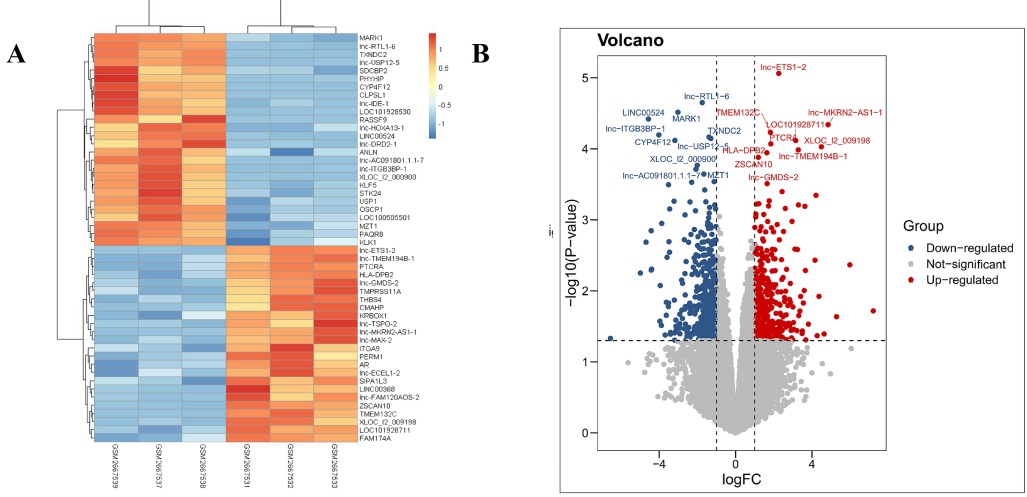

**Figure 1** **Heat map and volcanic plot of DEGs comparing patients with PDL1-positive and PDL1-negative lung adenocarcinoma.** (A) Heat map of DEGs. (B) Volcanic plot of DEGs. Red represents high expression levels in the PDL1-positive group, whereas blue represents high expression levels in the PDL1-negative group. Gray represents no difference in expression levels between the two groups.

revealed that the enriched pathways were mainly those involved in DNA replication, cell cycle, and mismatch repair (Table 2). The detailed results are shown in Table 2. The top 10 GO terms and KEGG enriched pathways were mapped using the Hmisc and ggplot2 packages. The results are shown in Fig. 2.

## GSEA enrichment analysis and validation
The results were validated through GSEA analysis after all the DEGs were considered. The GSEA results also provided the KEGG results. However, a new pathway enrichment was observed in the PDL1-positive group, namely the T-cell receptor pathway, which was not observed in KEGG results of DEGs (Fig. 3).

## Construction of PPI network and analysis of hub genes
The STRING database consists of known and predicted PPIs. The interactions include direct (physical) and indirect (functional) associations. To explore the expression relationships between the DEGs, a PPI network was constructed in our study. Our results showed that the PPI network consisted of 570 nodes and 2937 edges (Fig. 4A). The average node degree was 10.3 and the average local clustering coefficient was 0.404 (PPI enrichment $p$ value: <1.0e−16). In addition, the MNC algorithm was used to screen and identify hub genes that might be key genes with crucial regulatory functions. The top 10 hub genes, BUB1B, CDC45, BUB1, TTK, BRCA1, TOP2A, NDC80, RFC4, MCM2, and DTL, were identified in our study (Fig. 4B). The results were ranked using the MNC method, as shown in Table 3.

## Analysis of the mechanism of hub genes in the TCGA database
To prevent the limitations caused by using a single GEO database, our results were verified using the TCGA and GTEx databases. First, comparative analysis of hub gene expression

**Table 1  Top 10 GO terms.**

| Category | Term | Count | % | P-Value | FDR |
|---|---|---|---|---|---|
| PDL1 positive group | | | | | |
| GOTERM_BP_FAT | Cell adhesion | 21 | 8.713692946 | 2.88E−04 | 0.469256078 |
| GOTERM_BP_FAT | Biological adhesion | 21 | 8.713692946 | 2.94E−04 | 0.478025436 |
| GOTERM_BP_FAT | Cell–cell adhesion | 12 | 4.979253112 | 5.17E−04 | 0.841085346 |
| GOTERM_BP_FAT | Regulation of cell motion | 10 | 4.149377593 | 5.50E−04 | 0.894179746 |
| GOTERM_BP_FAT | MAPKKK cascade | 9 | 3.734439834 | 0.00173479 | 2.79437728 |
| GOTERM_CC_FAT | Plasma membrane | 81 | 33.60995851 | 1.28E−08 | 1.64E−05 |
| GOTERM_CC_FAT | Intrinsic to plasma membrane | 33 | 13.69294606 | 2.53E−05 | 0.032468756 |
| GOTERM_CC_FAT | Intrinsic to membrane | 93 | 38.58921162 | 5.00E−05 | 0.064061563 |
| GOTERM_CC_FAT | Integral to plasma membrane | 31 | 12.86307054 | 1.03E−04 | 0.132137329 |
| GOTERM_CC_FAT | Membrane raft | 8 | 3.319502075 | 0.001913071 | 2.423429637 |
| GOTERM_MF_FAT | Cadmium ion binding | 3 | 1.244813278 | 0.006756465 | 8.971494974 |
| GOTERM_MF_FAT | Sugar binding | 8 | 3.319502075 | 0.012297355 | 15.76520126 |
| GOTERM_MF_FAT | Carbohydrate binding | 11 | 4.564315353 | 0.01503554 | 18.94591653 |
| GOTERM_MF_FAT | Calcium ion binding | 20 | 8.298755187 | 0.024264811 | 28.86459093 |
| GOTERM_MF_FAT | Lipid binding | 12 | 4.979253112 | 0.02867722 | 33.1973471 |
| PDL1 negative group | | | | | |
| GOTERM_BP_DIRECT | Cell division | 34 | 8.629441624 | 4.29E−15 | 7.22E−12 |
| GOTERM_BP_DIRECT | DNA replication | 22 | 5.583756345 | 4.68E−13 | 7.79E−10 |
| GOTERM_BP_DIRECT | Mitotic nuclear division | 26 | 6.598984772 | 2.39E−12 | 3.98E−09 |
| GOTERM_BP_DIRECT | Chromosome segregation | 14 | 3.553299492 | 1.75E−10 | 0.000000291 |
| GOTERM_BP_DIRECT | G1/S transition of mitotic cell cycle | 15 | 3.807106599 | 3.32E−09 | 0.00000554 |
| GOTERM_CC_DIRECT | Nucleoplasm | 89 | 22.58883249 | 1.26E−09 | 0.00000169 |
| GOTERM_CC_DIRECT | Condensed chromosome kinetochore | 14 | 3.553299492 | 2.29E−09 | 0.00000306 |
| GOTERM_CC_DIRECT | Nucleus | 136 | 34.5177665 | 6.77E−08 | 0.0000906 |
| GOTERM_CC_DIRECT | Chromosome, centromeric region | 10 | 2.538071066 | 0.00000043 | 0.000576 |
| GOTERM_CC_DIRECT | Kinetochore | 11 | 2.791878173 | 0.00000104 | 0.001390211 |
| GOTERM_MF_DIRECT | DNA binding | 52 | 13.19796954 | 0.000106 | 0.150583673 |
| GOTERM_MF_DIRECT | Protein binding | 190 | 48.22335025 | 0.000124 | 0.175586276 |
| GOTERM_MF_DIRECT | Chromatin binding | 19 | 4.822335025 | 0.000252 | 0.356042613 |
| GOTERM_MF_DIRECT | DNA helicase activity | 5 | 1.269035533 | 0.000804 | 1.132060211 |
| GOTERM_MF_DIRECT | ATP binding | 43 | 10.91370558 | 0.002267608 | 3.161395589 |

in lung adenocarcinoma and adjacent normal tissue was performed in our study, and our results showed that the expression levels of BUB1B, CDC45, BUB1, TTK, TOP2A, NDC80, MCM2, and DTL were significantly higher in lung adenocarcinoma tissues than in adjacent normal lung tissues ($P < 0.01$). However, although the expression levels of BRCA1 and RFC4 were higher in lung adenocarcinoma tissues than in adjacent normal tissues, the differences were not significant (Fig. 5).

In addition, the relationship between hub genes and pathological staging were analyzed. The results showed that BUB1B ($F$ value $= 5.22$, $p = 0.00148$), CDC45 ($F$ value $= 2.86$, $P = 0.0364$), BUB1 ($F$ value $= 5.22$, $P = 0.00149$), TTK ($F$ value $= 5.06$, $P = 0.00185$), BRCA1 ($F$ value $= 4.9$, $P = 0.0023$), TOP2A ($F$ value $= 2.88$, $P = 0.0354$), NDC80 ($F$

**Table 2   Top 10 KEGG pathway enrichment results.**

| Category | Term | Count | % | *P*-Value | Genes | FDR |
|---|---|---|---|---|---|---|
| PDL1 positive group | | | | | | |
| KEGG_PATHWAY | Arrhythmogenic right ventricular cardiomyopathy (ARVC) | 7 | 2.904564315 | 0.001300189 | LAMA2, ITGA9, CACNA2D1, RYR2, ITGA10, CACNA2D3, CTNNA3 | 1.414211029 |
| KEGG_PATHWAY | Cell adhesion molecules (CAMs) | 7 | 2.904564315 | 0.019094239 | NCAM2, ITGA9, SELP, CDH15, CD22, CLDN22, HLA-DQA1 | 19.02707303 |
| KEGG_PATHWAY | PPAR signaling pathway | 5 | 2.074688797 | 0.024587412 | LPL, SLC27A1, OLR1, FABP3, ANGPTL4 | 23.85529626 |
| KEGG_PATHWAY | Complement and coagulation cascades | 5 | 2.074688797 | 0.024587412 | KNG1, CD55, CR2, F3, CFD | 23.85529626 |
| KEGG_PATHWAY | ECM-receptor interaction | 5 | 2.074688797 | 0.045850772 | LAMA2, ITGA9, ITGA10, CHAD, THBS4 | 40.17953232 |
| PDL1 negative group | | | | | | |
| KEGG_PATHWAY | DNA replication | 10 | 2.538071066 | 4.44E−09 | RFC3, RFC4, POLD2, PCNA, POLA1, MCM2, MCM3, MCM5, CM6, RPA3 | 0.00000536 |
| KEGG_PATHWAY | Cell cycle | 15 | 3.807106599 | 1.25E−08 | E2F2, SKP2, TTK, SMAD2, MCM2, MCM3, MCM5, MCM6, CCNE2, CDC45, CDKN2A, PCNA, BUB1, BUB1B, ORC1 | 0.0000151 |
| KEGG_PATHWAY | Mismatch repair | 5 | 1.269035533 | 0.000497 | RFC3, RFC4, POLD2, PCNA, RPA3 | 0.598123244 |
| KEGG_PATHWAY | Nucleotide excision repair | 6 | 1.52284264 | 0.001008028 | RFC3, RFC4, POLD2, PCNA, GTF2H4, RPA3 | 1.210410161 |
| KEGG_PATHWAY | p53 signaling pathway | 6 | 1.52284264 | 0.004883612 | CCNE2, CDKN2A, SERPINB5, RPRM, PERP, GTSE1 | 5.740021178 |

value $= 3.58$, $P = 0.014$), MCM2 ($F$ value $= 2.78$, $P = 0.0407$), and DTL ($F$ value $= 5.97$, $P = 0.000535$) were positively correlated with pathological staging, and the differences were statistically significant. However, regarding the correlation between RFC4 and pathological staging, although RFC4 was positively correlated with pathological staging, the expression of RFC4 did not differ significantly between the two groups ($F$ value $= 1.82$, $P = 0.142$; Fig. 6).

Finally, the relationship between the hub genes and prognosis was analyzed. The results showed that patients with lung adenocarcinoma with low expression levels of BUB1B (Log rank $P = 3.8e −05$, P (HR) $= 5.1e −05$ ), CDC45 (log rank $P = 0.0032$, p (HR) $= 0.0035$), BUB1 (log rank $P = 0.0024$, p (HR) $= 0.0026$ ), TTK [log rank $P = 0.00029$, P (HR) $= 0.00034$], BRCA1 [log rank $P = 0.0035$, P (HR) $= 0.0038$], TOP2A [log rank $P = 0.011$, P (HR) $= 0.012$], NDC80 [log rank $P = 0.0027$, P (HR) $= 0.003$], MCM2 [log rank $P = 0.02$, P (HR) $= 0.021$], and DTL [log rank $P = 0.0016$, P (HR) $= 0.0018$] had significantly higher overall survival than those with high gene expression levels ($P < 0.05$). However,

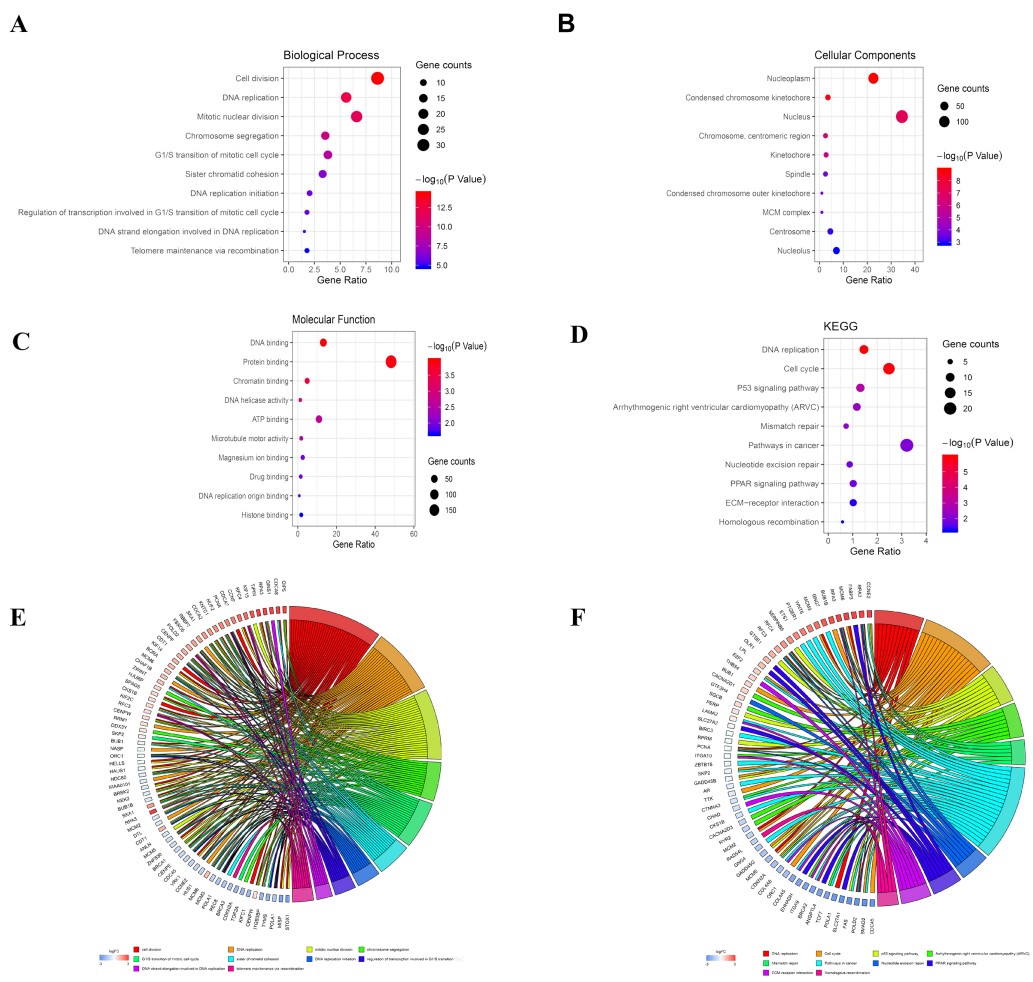

**Figure 2** **Bubble mapping of the top 10 GO terms and KEGG pathway enrichment analysis data of DEGs.** (A) GO analysis of DEGs in biological process. (B) GO analysis of DEGs in cellular components. (C) GO analysis of DEGs in terms of molecular function. (D) KEGG enrichment analysis of DEGs. A high gene ratio represents a high level of enrichment. The size of the dot indicates the number of target genes in the pathway and the color of the dot reflects the *p* value range. (E) GO Chord plot of DEGs. (F) KEGG Chord plot of DEGs.

no significant correlation was observed between RFC4 gene expression level and overall survival (log rank $P = 0.059$; Fig. 7).

## Analysis of the mechanism of hub genes in the TCGA database

Furthermore, analysis of the lung adenocarcinoma was performed using the database TCGA. The results showed that the hub genes had different mutation frequencies, ranging from 1% to 5%. For example, BUB1B (4%), CDC45 (1.9%), and DTL (5%) (Fig. 8A). Moreover, the potential hub genes exhibited different mutation forms in lung adenocarcinoma, which may facilitate further exploration of the function of these hub genes in lung adenocarcinoma. For example, BUB1B mainly exhibited deep deletion, whereas *BRCA1* mainly exhibited amplification (Fig. 8B). However, these different mutant forms

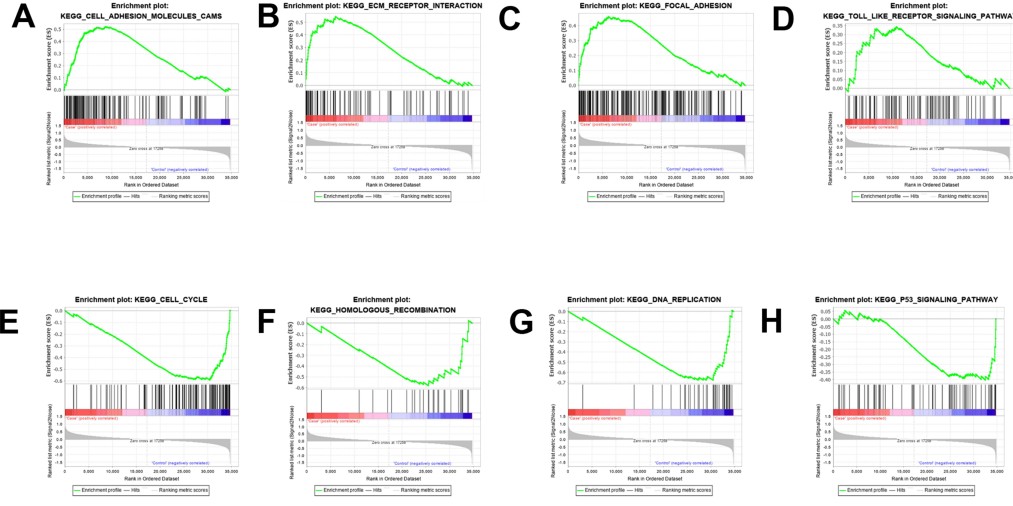

**Figure 3  GSEA enrichment analysis and validation.** (A, B, C, D) GSEA enrichment analysis in the PDL1-positive group. (E, F, G, H) GSEA enrichment analysis in the PDL1-negative group.

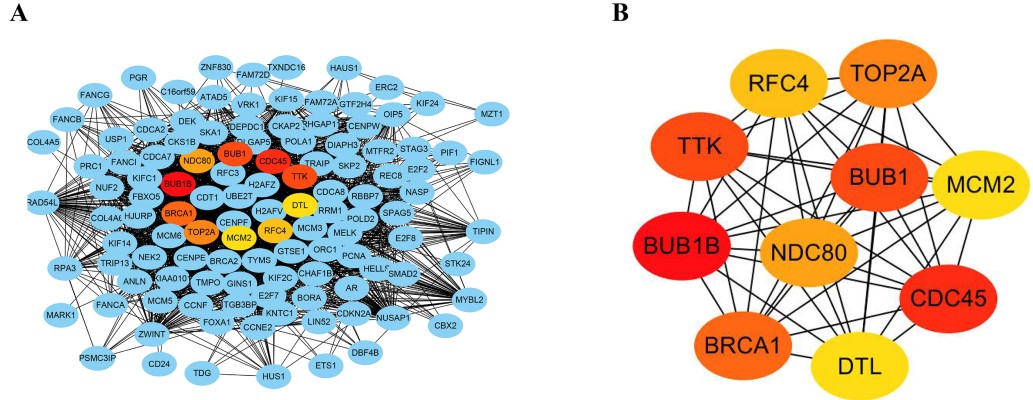

**Figure 4  Construction of PPI networks and analysis of hub genes.** (A) The PPI network was constructed using Cytoscape. (B) The top 10 hub genes were screened using the MNC algorithm. The color of the hub gene holds importance according to the MNC algorithm.

of hub genes were involved in different signaling pathways influencing the development of lung adenocarcinoma; for example, *BRCA1* was mainly involved in the cell cycle and DNA damage response pathways (Figs. 9A and 9B).

## DISCUSSION

Lung adenocarcinoma is a highly heterogeneous type of cancer, and its individualized treatment has attracted considerable attention. An increasing number of studies have recently indicated that immunotherapy, especially anti-PD-1 therapy, is a promising strategy for treating lung adenocarcinoma. Some anti-PD-1 antibodies have been successfully commercialized and approved as first- and second-line immunotherapy

**Table 3  Top 10 hub genes ranked using the MNC method.**

| Rank | Name | Score |
| --- | --- | --- |
| 1 | BUB1B | 81 |
| 2 | CDC45 | 79 |
| 3 | BUB1 | 78 |
| 3 | TTK | 78 |
| 5 | BRCA1 | 75 |
| 6 | TOP2A | 73 |
| 7 | NDC80 | 72 |
| 8 | RFC4 | 71 |
| 9 | MCM2 | 69 |
| 9 | DTL | 69 |

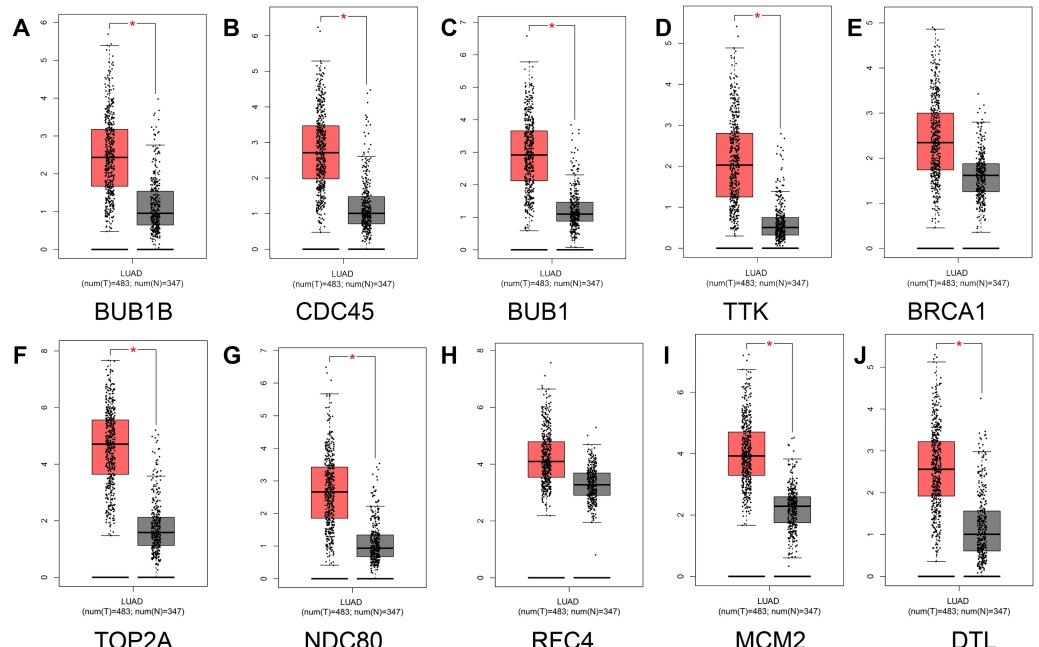

**Figure 5  Analysis of hub gene expression in lung adenocarcinoma.** The red and gray boxes represent lung adenocarcinoma and adjacent normal tissue, respectively. (A) BUB1B; (B) CDC45; (C) BUB1, (D) TTK; (E) BRCA1; (F) TOP2A; (G) NDC80; (H) RFC4; (I) MCM2; and (J) DTL.

options for advanced NSCLC in the past few years. Nevertheless, the overall response rate for such immunotherapy is only approximately 20% (*Greillier, Tomasini & Barlesi, 2018*), and the efficacy of such immunotherapy is affected by PDL1 expression. In particular, PDL1 expression has been proposed as a predictive biomarker, suggesting that it plays a major role in immune regulation in lung adenocarcinoma. Therefore, understanding the effects of PDL1 expression on the biological behavior and the efficacy of immunotherapy in lung adenocarcinoma is crucial. However, the use of IHC alone to assess PDL1 expression cannot provide a complete explanation for the molecular mechanism and enriched

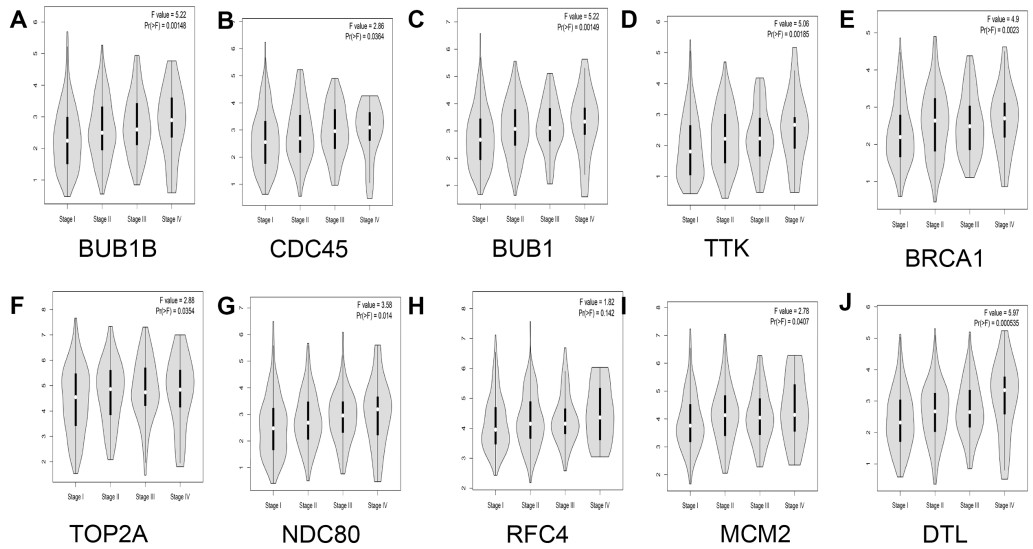

**Figure 6** **Analysis of the relationship between hub genes and pathological staging in lung adenocarcinoma.** (A) BUB1B; (B) CDC45; (C) BUB1; (D) TTK; (E) BRCA1; (F) TOP2A; (G) NDC80; (H) RFC4; (I) MCM2; and (J) DTL.

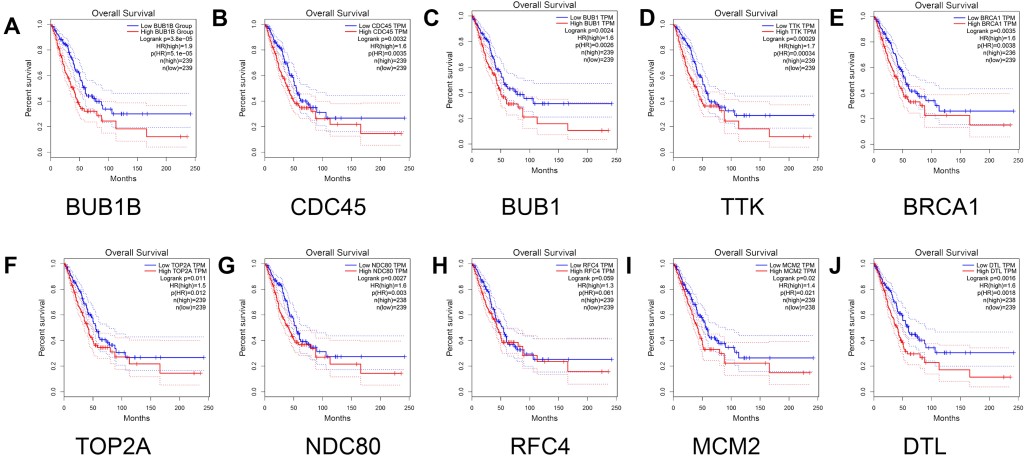

**Figure 7** **Survival analysis for patients with lung adenocarcinoma in relation to the expression of hub genes.** (A) BUB1B; (B) CDC45; (C) BUB1; (D) TTK; (E) BRCA1; (F) TOP2A; (G) NDC80; (H) RFC4; (I) MCM2; and (J) DTL. Red and blue represent high and low expression levels of hub genes, respectively.

pathways underlying the efficacy of immunotherapy. With the development of next-generation sequencing technology and bioinformatics, a novel strategic solution to this problem may be identified (*Morganti et al., 2019*). Furthermore, previous studies have used bioinformatic analysis to explore the core genes of lung adenocarcinoma and their malignant transformation mechanisms (*Yuan et al., 2017*; *Yeh et al., 2019*). However, no research has been conducted on PDL1-positive and PDL1-negative patients. Understanding

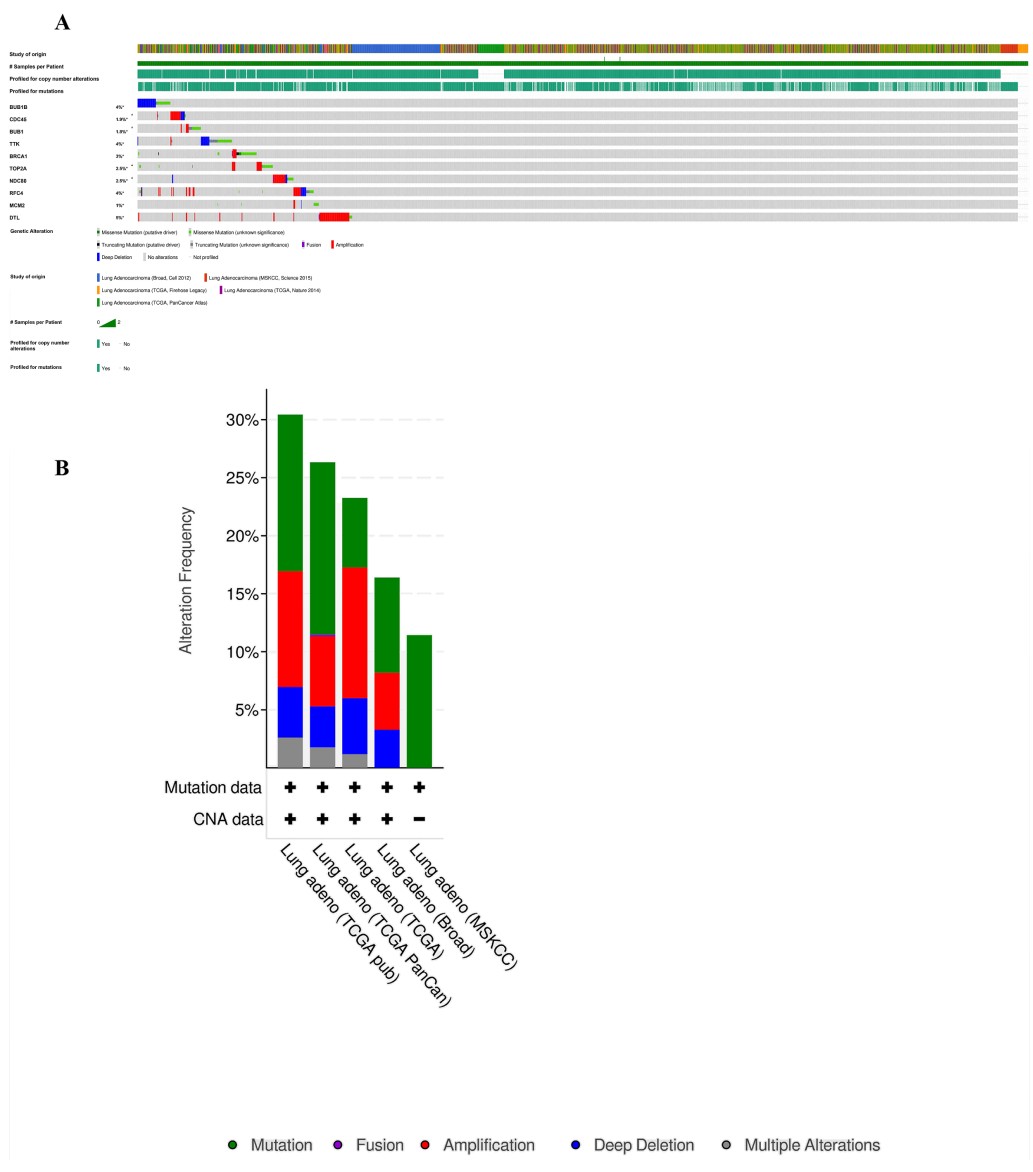

**Figure 8  Analysis of mutation characteristics of hub genes.** (A) Matrix heatmap shows genomic alterations of hub genes in five lung data sets (Broad, Cell (2012); MSKCC, Science (2015); TCGA, Firehose Legacy; TCGA, Nature (2014); and TCGA, PanCancer Atlas). (B) The alteration frequencies of hub genes across five studies on lung adenocarcinoma.

DEGs and differences in the biological processes and enrichment pathways between PDL1-positive and PDL1-negative patients is crucial. Hence, by using gene bioinformatics analysis, we explored the differences in gene expression profiles and enrichment pathways between the two groups of patients and identified the potential key biomarkers that could be used for predicting disease prognosis in patients. The elucidation of the regulatory mechanism of PDL1 is crucial to improve precision immunotherapy for lung adenocarcinoma.
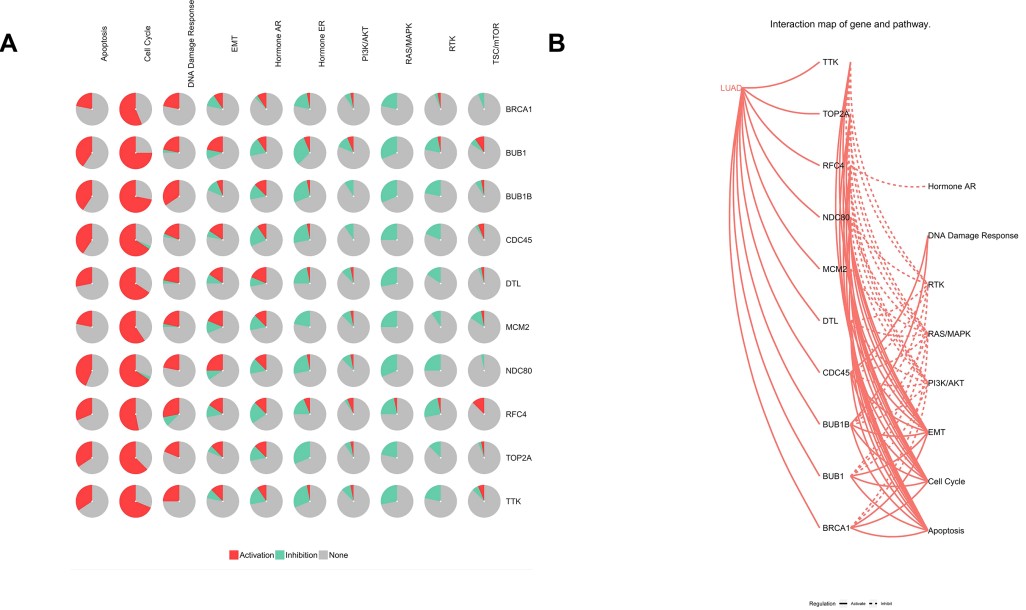

**Figure 9  Analysis of pathways of Hub genes.** (A) Pie chart of hub genes involved in pathways. (B) Interaction map of hub genes and pathways in lung adenocarcinoma.

Our study demonstrated that even in patients with lung adenocarcinoma, differences in gene expression were observed between those in the PDL1-positive and PDL1-negative group, thus suggesting that these two types of patients should be treated differently.

The important findings in our study in the GO terms and KEGG pathway analysis were the differences between the PDL1-positive and PDL1-negative groups of patients with lung adenocarcinoma. Our results reveal that the PDL1-positive group mainly demonstrated enrichment of adhesion-related biological processes and the PPAR signaling pathway. Adhesion-related biological processes have been previously reported to be associated with invasion and metastasis in patients with lung adenocarcinoma (*Stevens et al., 2017*; *Alonso-Nocelo et al., 2018*), resulting in relatively high levels of organ and tissue metastases and poor prognosis. Real-world studies have revealed similar results (*Wang et al., 2015*; *Okita et al., 2017*). Previous studies have reported that PPAR, an anti-inflammatory molecule with a potent, could promote tumor proliferation, angiogenesis, inflammation, and metastasis in lung adenocarcinoma (*Reka et al., 2011*; *Ammu et al., 2019*), suggesting that it is a vital and potential therapeutic target for lung adenocarcinoma. Moreover, *Lv & Wang (2015)* discovered that the PPAR signaling pathway played a major role in malignant transformation of cells among nonsmoking patients with lung adenocarcinoma. These results are consistent with our findings. Although several studies have suggested the relationship between PPAR and lung adenocarcinoma, the role of PPAR in immunotherapy for lung adenocarcinoma has yet to be reported. Moreover, recent studies have demonstrated that PPAR could enhance indoleamine 2,3-dioxgenase-1 (IDO) activity and promote the generation of regulatory T cells in melanoma; hence, the use of PPAR inhibitors could enhance cancer immunotherapy (*Poupot et al., 2014*). On

the basis of our results along with the results of these studies, we hypothesized that the PPAR pathway might play a crucial role in immunotherapy for lung adenocarcinoma. A comprehensive analysis of our results indicated that although PDL1 expression was associated with the immune response pathway, it might have potential to contribute to invasion and metastasis in patients with PDL1-positive lung adenocarcinoma.

Conversely, the results of the PDL1-negative group mainly revealed enrichment of division-related biological processes and repair systems pathways, which indicated that these pathways might be able to repair defects in apoptosis or recombination to reduce the production of neoantigens. Some clinical studies have shown that PDL1-negative patients tend to have lower tumor mutation burdens or fewer neoantigens than PDL1-positive patients (*Xia et al., 2017*; *Chan et al., 2019*), which might be related to its involvement in various repair systems. Further research must be conducted to confirm our conclusions.

In addition, the GSEA results were similar to those for the KEGG pathway analysis. More importantly, a notable and novel finding was reported in the PDL1-positive group, namely that related to the T-cell receptor pathway, which has been proved to be crucial for achieving a favorable immunotherapeutic response in NSCLC (*Van De Ven & Borst, 2015*). This finding suggests that PDL1 might be involved in the regulation of the T-cell pathway; therefore, understanding the effects of PDL1 expression on tumor microenvironmental immune cells and mediation of T-cell pathways is necessary.

In our study, 10 hub genes screened in the GEO database were verified and further analyzed using the TCGA and GTEx databases. Our study further analyzed the prognostic value and possible mechanism of hub genes in lung adenocarcinoma considering their importance.

Several hub genes, such as BUB1B, CDC45, BUB1, TTK, TOP2A, MCM2, NDC80, and DTL, have been reported to be associated with poor prognosis in patients with lung adenocarcinoma (*Hayama et al., 2006*; *He et al., 2019*; *Liu et al., 2017*; *Perez-Pea et al., 2017*; *Song et al., 2018*; *Sun et al., 2020*). Our findings corroborate these results. Another major finding of our study is that BRCA1 is a predictor in patients with lung adenocarcinoma.

BRCA1 encodes a nuclear phosphoprotein that helps maintain genomic stability. It is involved in biological processes such as cell cycle regulation, replication, and mitotic spindle assembly. Previous studies have suggested that BRCA1 overexpression regulates drug response in chemotherapy, is related to the efficacy of EGFR-TKIs, and is prevalent in patients with NSCLC with early disease onset (*Reguart et al., 2008*; *Sun et al., 2018*). Recent studies have analyzed the potential prognostic role of BRCA1 in early-stage NSCLC, suggesting that BRCA1 is a predictor of survival in only stage III NSCLC (*Hu et al., 2019*). Although several studies have demonstrated the role of BRCA1 in NSCLC, the prognostic value and possible mechanism of BRCA1 in lung adenocarcinoma remain unclear. Our study results suggest that BRCA1 overexpression was associated with poor prognosis in lung adenocarcinoma. In particular, our results indicate that BRCA1 was mainly involved in cell cycle and DNA damage responses in lung adenocarcinoma, which might produce new antigens and enhance immune responses. BRCA1 mutation was also reported to be associated with tumor neoantigen production, immune cell invasion, and PDL1 expression

in ovarian cancer (*Strickland et al., 2016*). Overall, these findings suggest that BRCA1 plays a key role in the immunomodulatory pathway of lung adenocarcinoma.

RFC4, also named activator 1, is a protein complex consisting of five subunits measuring 140, 40, 38, 37, and 36 kD. Telomere C-strand (Lagging Strand) Synthesis and E2F-mediated regulation of DNA replication are some of its related pathways. A previously conduced weighted gene co-expression network analysis revealed that it was a prognosis-related biomarker in lung adenocarcinoma (*Yi et al., 2020*). However, our GEPIA survival analysis showed that RFC4 overexpression was positively correlated with pathological staging but was not associated with poor prognosis in patients with lung adenocarcinoma. Possible reasons for these inconsistent findings are differences in sample selection strategies and algorithms. Accordingly, the prognostic value of RFC4 for lung adenocarcinoma warrants further study.

The present study has several limitations. First, only a single GSE was searched and analyzed in the GEO database. To prevent bias resulting from the analysis of a single data set, we adopted an integrated bioinformatic analysis approach. Second, the main objective of this study was to explore the influence of PDL1 expression on related gene expression profiles in patients with lung adenocarcinoma. Hence, this study was divided into expression in PDL1-positive and PDL1-negative groups; however, no further distinction was made in the PDL1-positive group between patients with high (50%) and low (<1%) levels of PDL1 expression. Different PDL1 expression levels might have caused differences in the expression profiles of genes, which warrants further confirmation. Third, some key genes have been studied in various tumors, including non–small-cell lung cancer. However, this study identified core genes that might be related to or regulated by the expression of PDL1 in lung adenocarcinoma, which will assist in the further discovery of mechanisms for designing novel immunotherapeutic options.

This study revealed the DEGs and different biological pathways between PDL1-positive and PDL1-negative patients with lung adenocarcinoma, speculating that these two types of patients might have different subtypes of lung adenocarcinoma. The identification and verification of hub genes through integrated bioinformatic analysis revealed that they related to immune response pathways and prognosis in patients with lung adenocarcinoma. In particular, they had different mutations and were involved in different pathways in lung adenocarcinoma.

## CONCLUSIONS

PDL1-positive lung adenocarcinomaand PDL1-negative lung adenocarcinoma might be different subtypes of lung adenocarcinoma. Potential hub genes might be involved in PDL1 regulatory pathways, and further research is warranted to reveal new mechanisms underlying the regulation of PDL1 expression. This could be of great significance for precision immunotherapy for lung adenocarcinoma.

### Funding

The authors received no funding for this work.

### Competing Interests

The authors declare there are no competing interests.

### Author Contributions

- Xiaoguang Qi conceived and designed the experiments, performed the experiments, analyzed the data, prepared figures and/or tables, authored or reviewed drafts of the paper, and approved the final draft.
- Chunyan Qi performed the experiments, prepared figures and/or tables, authored or reviewed drafts of the paper, and approved the final draft.
- Xindan Kang performed the experiments, analyzed the data, prepared figures and/or tables, and approved the final draft.
- Yi Hu and Weidong Han conceived and designed the experiments, authored or reviewed drafts of the paper, and approved the final draft.

### Data Availability

Data is available at NCBI GEO: GSE99995.

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
