# Peer review of "Identification of candidate genes and prognostic value analysis in patients with PDL1-positive and PDL1-negative lung adenocarcinoma"

_PeerJ, doi:10.7717/peerj.9362_

## Round 0.1 · original submission · Major Revisions

Both the reviewers have pointed out major modifications in the manuscript. Please address all the raised concerned in the revised manuscript.

Reviewer 1 ·

Basic reporting

In this manuscript, the authors identify the DEG and different biological pathways that are involved in the patients of PDL-1 positive and negative lung adenocarcinoma. Furthermore, the authors report the key genes that can be used as a biomarker for lung cancers.

There is no sufficient background provided about the role of PDL-1 in lung cancer, hard to understand the purpose of the study. I think the authors should include more background and details from the previous reports in the "Introduction" section for better understanding to broad interest readers. Re-write the abstract and introduction section.

The authors follow the journal's guidelines for article structure and figures and table preparation.

Experimental design

The data presented in the manuscript are clear and corroborate the conclusion made by the authors.

The major issue with the experimental design is related to a limited (3 patients) number of the dataset in each group which I think is very low and seems insufficient to make any conclusive interpretation. Please comment.

However, I would like to provide a few suggestions that will improve the quality of the manuscript:

1. Re-write the materials and methods section (provide more details); seems incomplete
2. In the figure section, there are a few minor errors:
a) Figure 1 legend, Typo-error: change "GEGs" to "DEGs"
b) Figure 1B legend: There are no black and green color dots in Figure 1B. Change "green" to "blue" and "black" to "grey"
3) In Figures 5, 6, and 7: Write the name of the gene in the inset of the boxes will make it very easy for readers to follow.

Validity of the findings

No comments

Additional comments

There is already existing literature that highlights the role of BRCA1 and NDC80 in lung cancers and the authors fail to cite any article which supports their findings. I think the authors should re-write the "Introduction and Discussion" section with proper literature citation.
Cite this article:
1. Potential therapeutic targets of the nuclear division cycle 80 (NDC80) complexes genes in lung adenocarcinoma, Cancer 2020; 11(10):2921-2934. doi:10.7150/jca.41834.
2. Are we ready to use biomarkers for staging, prognosis and treatment selection in early-stage non-small-cell lung cancer? doi: 10.3978/j.issn.2218-6751.2013.03.06

Reviewer 2 ·

Basic reporting

In this work, the authors used bioinformatics tools to analyze gene expression dataset between PDL1-positive and PDL1-negative patients with lung adenocarcinoma and suggest BRAC1 and NDC80 as new prognostic indicators for patients with lung adenocarcinoma.

Sufficient background information not provided as further elaborated in general comments for the authors.

Experimental design

The aim of the study is well-defined and the methods used are appropriate.

Validity of the findings

Overall conclusion are supported by the results shown. However, the novelty of the findings is not assessed as further elaborated in general comments for the authors.

Additional comments

1) The introduction needs improvement as sufficient background covering previous bioinformatics work identifying hub genes and potential biomarkers for lung cancer (to mention a few, DOIs: 10.3892/ol.2018.8882, 10.3892/ol.2019.10796 etc.) is not provided to understand the significance and novelty of the work.
2) In the discussion authors should elaborate more on the novelty of the hub genes identified in this work as previous literature have already associated the hub genes identified here with lung adenocarcinoma. For example, on lines 245-246, “…Our study revealed that BUB1B, BUB1, MCM2, and DTL might be new potential oncogenes.…”. However previous works have associated these genes with lung adenocarcinoma (DOIs: 10.18632/genesandcancer.53, 10.3892/ol.2019.10796, 10.1038/s41598-017-13440-x, 10.1038/s41598-017-00512-1). Additionally, on lines 263-264, “…Moreover, we identified some new hub genes, such as BRCA1 and NDC80, which were associated with prognosis in patients with lung adenocarcinoma.…”. Again, both BRAC1 and NDC80 have been associated with lung adenocarcinoma in previous studies (DOIs: 10.3816/CLC.2008.n.048, 10.20892/j.issn.2095-3941.2018.0506, 10.1158/0008-5472.CAN-06-2137).
3) One limitation of the analysis is in the dataset used which seems to consists of data from only 3 patients in each of the two groups (PDL1-postive and PDL1-negative). On line 77, “…GSE99995 was retrieved.…”. Was this the only dataset retrieved? If not, why was this dataset selected in particular. Also, in the methods section authors should add the criteria on which the patients were divided into two groups.
4) Line 130, “…Our results showed that the two groups comprised 34729 genes…”. Authors should write the names of the two groups.
5) Figure 2 seems to present data only for the PDL1-negative group, this should be identified in the text and figure legend.
6) There are typos at some places in the manuscript. For example, “GEGs” should be “DEGs” and “green” should be “blue” in Figure 1 legend, “he” should be changed to “the” on line 281, etc.

---

## Round 0.2 · Minor Revisions

As pointed out by the reviewer, please correct all the spelling and grammatical mistakes in this manuscript before re-submitting.

Reviewer 1 ·

Basic reporting

No comment

Experimental design

No comment

Validity of the findings

No comment

Additional comments

The revised manuscript is now very clear with a detailed description of the methods. The authors have now addressed all the critical issues raised by both reviewers.
However, the manuscript needs English language editing in the Introduction and Discussion sections. I think the authors can use language editing services provided by Peer J.

Reviewer 2 ·

Basic reporting

NA

Experimental design

NA

Validity of the findings

NA

Additional comments

The authors have addressed all of my concerns. However, there are spelling and grammatical mistakes at several places in the manuscript that needs correction. To give a few examples: "differences expression genes" on line 26, "studies have shown indicated" on line 82, etc. Also, fix lines 97-99.
I would recommend publishing the manuscript provided authors proofread and correct for mistakes in English.

---

## Round 0.3 · accepted · Accept

Authors have addressed all the question raised by the reviewers. Manuscript is ready for publication.

Reviewer 1 ·

Basic reporting

No comment

Experimental design

No comment

Validity of the findings

No comment

Additional comments

No comment

Reviewer 2 ·

Basic reporting

NA

Experimental design

NA

Validity of the findings

NA

Additional comments

Authors have performed suggested changes and the manuscript is now suitable for publication.